# Thermal adaptation best explains Bergmann's and Allen's Rules across ecologically diverse shorebirds

Alexandra McQueen [1], Marcel Klaassen [2], Glenn J. Tattersall [3], Robyn Atkinson[4], Roz Jessop[4], Chris J. Hassell[5], Maureen Christie[6], Victorian Wader Study Group*, Australasian Wader Studies Group* & Matthew R. E. Symonds [1] ✉

Bergmann's and Allen's rules state that endotherms should be larger and have shorter appendages in cooler climates. However, the drivers of these rules are not clear. Both rules could be explained by adaptation for improved thermoregulation, including plastic responses to temperature in early life. Non-thermal explanations are also plausible as climate impacts other factors that influence size and shape, including starvation risk, predation risk, and foraging ecology. We assess the potential drivers of Bergmann's and Allen's rules in 30 shorebird species using extensive field data (>200,000 observations). We show birds in hot, tropical northern Australia have longer bills and smaller bodies than conspecifics in temperate, southern Australia, conforming with both ecogeographical rules. This pattern is consistent across ecologically diverse species, including migratory birds that spend early life in the Arctic. Our findings best support the hypothesis that thermoregulatory adaptation to warm climates drives latitudinal patterns in shorebird size and shape.

Ecogeographical rules describe global variation in animal form and function. These classic rules are subject to renewed attention as ecologists attempt to predict the consequences of climate change for species ecology and survival[1–3]. Two rules explain variation in the size and shape of endothermic animals. Bergmann's rule states that animals are larger in colder climates[4]. Meanwhile, Allen's rule states that animals have shorter appendages—such as ears, tails and limbs—in colder climates[5]. Both rules have received widespread empirical support[6–11]. However, there are exceptions to the rules, including studies that question their generality[12–16], and the drivers of geographic variation in animal size and shape are not well understood[17,18].

Bergmann and Allen first predicted latitudinal clines in animal size and shape based on thermal adaptation to different climates ('thermoregulatory hypothesis'[4,5,19]). In cold climates, larger body sizes and shorter appendages reduce the surface area available for heat loss, allowing animals to conserve heat. By the same logic, smaller bodies and longer appendages are advantageous in warm climates due to the increased relative surface area available for heat loss. Many studies cite the former explanation—that it is better to be larger and have shorter appendages where it is cold—as the crucial driver of latitudinal clines[20]. An emphasis on cool climate effects is supported by studies showing stronger effects of minimum winter temperatures than maximum summer temperatures on geographic patterns in size and shape[11,21–24]. However, warm temperatures can further impact animal size and shape, particularly in freshwater-limited coastal or desert environments, and humid environments, where there is limited capacity for

[1]Centre for Integrative Ecology, School of Life and Environmental Sciences, Deakin University, Burwood, VIC 3125, Australia. [2]Centre for Integrative Ecology, School of Life and Environmental Sciences, Deakin University, Geelong, VIC 3216, Australia. [3]Department of Biological Sciences, Brock University, 1812 Sir Isaac Brock Way, Saint Catharines, ON L2S 3A1, Canada. [4]BirdLife Australia, Carlton, VIC 3053, Australia. [5]Global Flyway Network, PO Box 3089 Broome, WA 6725, Australia. [6]Friends of Shorebirds SE, Carpenter Rocks, SA 5291, Australia. *Lists of authors and their affiliations appear at the end of the paper. ✉e-mail: matthew.symonds@deakin.edu.au

## BOX 1
# Birds use their bills for thermoregulation

Birds can dissipate heat via their bills because they are unfeathered (non-insulated) with a network of blood vessels close to the surface. In hot conditions, birds increase blood flow to their bill surface, increasing heat loss via the bill to as much as 400% of resting heat production[103]. This behaviour is particularly effective in long-billed species due to an increased surface area for heat loss[60], where a small increase in bill size can have a substantial impact on the capacity to dissipate heat. For example, song sparrows adapted to coastal dunes and salt marsh have bills with a 9-mm$^2$ greater surface area than song sparrows adapted to mesic habitats. This difference corresponds to a ~33% increase in heat loss via the bill and a predicted ~8% reduction in water loss[104]. Conversely, large bills can be disadvantageous in cold climates because blood flow to the bill cannot entirely be restricted, meaning some heat loss is inevitable[60]. To compensate for this, birds with longer bills spend more time with their bills tucked beneath their feathers in cold weather[105,106]. Such behaviour may come at the cost of reduced time spent foraging or scanning for predators, and shorter bills may therefore be selected for in cold conditions to minimise thermal or time costs. Although birds also lose heat via their legs, counter-current blood flow in the legs reduces the amount of body heat lost in cold and warm conditions, potentially making legs less subject to thermoregulatory trade-offs[103]. The adaptation of bird bills for thermoregulation could explain why many bird bills follow Allen's rule[9].

evaporative heat loss via panting[10,25–27]. The expectation that Allen's rule is further driven by thermal adaptation to warm climates is compelling where appendages are highly adapted to promote heat loss[28]. For example, longer bills are advantageous in hot conditions because birds increase blood flow to the bill's surface to dissipate heat (Box 1).

These adaptive responses to climate are often assumed to have a population-level evolutionary genetic basis. This idea is supported by patterns observed across species, and cross-fostering experiments that demonstrate a strong genetic component to morphology related to climate[29]. However, plastic responses to temperature in early life could also explain latitudinal clines in animal size and shape ('developmental plasticity hypothesis'[30]). Exposure to high temperatures during development can lead to smaller adult body size, potentially due to heat stress during growth, reduced parental provisioning and other parental effects[31–33]. In addition, high temperatures directly influence skeletal growth in early development, leading to longer limbs in birds and mammals[34,35] and longer bills in birds[36], although differences may be reversibly plastic and not persist to adulthood[37]. Plastic responses themselves can result from adaptive evolution to increase fitness in the prevailing conditions in later life, or non-adaptive, direct effects of temperature on growth and development[35,38].

Potential non-thermal drivers of Bergmann's and Allen's rules include latitudinal patterns in starvation risk, predation risk, migration distance and foraging ecology. The risk of starvation is higher in cold and seasonal environments and should favour larger animals with greater energy stores at high latitudes; larger animals may also be favoured at high latitudes if they consume larger prey or a greater variety of prey sizes ('starvation risk hypothesis'[39–43]). Meanwhile, starvation risk decreases, and predation risk generally increases closer to the tropics[44,45]. This should select for lower body mass in animals and shorter wings in birds for improved agility and escape capacity

('predation risk hypothesis'[40,46,47]). Latitudinal clines in body size could relate to differences in migration behaviour, where migratory birds that fly further distances are selected to be larger and have longer wings for increased flight efficiency[48,49], or differently sized age classes or sexes migrate to different locations. For example, strong selection pressure for males to arrive early at the breeding grounds may favour males flying shorter distances to closer non-breeding grounds, leading to latitudinal patterns in body size for sexually dimorphic species ('differential migration hypothesis'[50,51]). Finally, Allen's rule may be related to latitudinal patterns in food size or foraging behaviour[12,52]. For example, larger insects in tropical regions could select for longer bills in insectivorous birds for improved handling of larger prey[53]. Likewise, shorebirds in tropical regions could have evolved longer bills because benthic invertebrates are buried deeper beneath the surface in warmer climates ('foraging ecology hypothesis'[54–56]).

Here we compare the bill and body size of shorebird populations in contrasting climates within Australia. We use extensive data (>200,000 observations) collected from 30 wild shorebird species over 46 years (1975–2021) by community scientists of the Victorian Wader Study Group (VWSG) and Australasian Wader Studies Group (AWSG). The shorebirds include long-distance migratory, nomadic or partly migratory, and resident species from five families with diverse morphological and life-history traits. This diversity offers an opportunity to test whether latitudinal patterns in body size and shape follow Bergmann's and Allen's Rules, and to distinguish between potential mechanisms of geographic variation in body size and shape (Box 2). Migratory shorebirds offer an opportunity to distinguish two key hypotheses for Bergmann's and Allen's rules: genetic adaptation for thermoregulation and developmental plasticity. Many migratory shorebirds undergo early growth and development on the cold, windswept Arctic tundra or Central-Asian cold desert breeding grounds before migrating to different locations in the southern hemisphere. Starting at similar breeding grounds, some individuals travel to hot, tropical northern Australia, while their conspecifics fly further south to relatively cool, temperate south-eastern Australia[57] (Fig. 1). If latitudinal patterns in size and shape are influenced by the genetic adaptation for thermoregulation in warm climates, we predict both smaller bodies and longer bills across ecologically diverse shorebirds in northern Australia (Box 2).

## Results

Across 30 species, northern Australian shorebirds have longer bills than southern Australian conspecifics. Bill length (relative to body size) in northern Australian populations is 1.76% greater than southern Australian populations (95% credible interval = 1.65–1.86%, $P < 0.001$, $N = 99,443$; the increase in absolute bill length = 1.12%, 95% CI = 1.01–1.23%, $P < 0.001$). Bills are significantly longer in northern populations for 15/18 migrant, 2/7 nomadic or partially migrant, and 4/5 resident species (Fig. 2A). Only two species—the red-necked avocet and sooty oystercatcher—have significantly shorter bills in northern, compared to southern Australia (Fig. 2A). Longer bills in northern populations are found across species with probing and visual foraging techniques (Fig. 3A). Resident species have greater differences in bill length between northern and southern populations compared to nomadic and migratory species (Fig. 3B).

Northern populations have smaller bodies than southern populations (wing length = −1.05%, 95% CI = −1.10, −1.01, $P < 0.001$, $N = 119,403$; mass = −6.69%, 95% CI = −6.83, −6.55, $P < 0.001$, $N = 202,647$). Northern populations also carry lower body stores (body mass controlling for wing length = −6.22%, 95% CI = −6.39, −6.05, $P < 0.001$, $N = 118,017$; Supplementary Fig. S2). Birds in northern populations have significantly shorter wings for 9/18 migrant, 4/7 nomadic or partially migrant, and 4/5 resident species (Fig. 2B). In contrast to most species, sharp-tailed sandpipers and great crested terns have significantly longer wings at northern sites (Fig. 2B). Birds in northern populations have a

## BOX 2

# Hypotheses underlying latitudinal gradients in bill length and body size

Predicted latitudinal differences in shorebird bill length and body size (estimated using wing length and body mass) according to six hypotheses. These hypotheses are not mutually exclusive and generate similar predictions; however, they differ in their capacity to explain latitudinal patterns in bill length and body size across shorebirds with diverse movement ecology. Arrows show predicted differences in shape and size for northern Australian relative to southern Australian (non-breeding) populations. Dashes indicate no direct predicted effects leading to differences between northern and southern Australian populations. Predicted differences are shown separately for migratory (M), nomadic or partly migratory (N) and resident (R) species (see Methods). The magnitude of the expected differences is shown as white = minor; grey = moderate; black = large. (1) 'Thermoregulatory hypothesis': if latitudinal differences in bird shape and size are explained by the genetic adaptation for thermoregulation, we predict northern Australian shorebirds will have longer bills and smaller bodies. We expect differences will be greatest for resident species because northern and southern Australian populations are exposed to very different climates year-round. (2) 'Developmental plasticity hypothesis': if latitudinal differences in shorebird size and shape are explained by plastic responses to climate conditions during early growth and development, we predict northern resident shorebirds to have longer bills and smaller bodies. Latitudinal differences should be absent for migratory birds because both northern and southern Australian populations breed at similar latitudes, and are exposed to cooler climates in the northern hemisphere during early

life. (3) 'Starvation risk hypothesis': if latitudinal differences are driven by greater starvation risk during winter in temperate climates, we predict shorebirds that experience cool winters in southern Australia will carry greater body stores than conspecifics in tropical northern Australia, leading to relatively smaller body size in northern populations. We do not predict an effect for nomadic species as these birds readily relocate to find food. (4) 'Predation risk hypothesis': if latitudinal differences are explained by greater predation risk at lower latitudes, we predict northern Australian shorebird populations to have shorter wings and smaller bodies than southern conspecifics. (5) 'Differential migration hypothesis': if latitudinal differences are driven by differential migration patterns—for example where smaller males of sexually dimorphic species fly shorter distances to northern Australian non-breeding grounds—we predict migratory shorebirds to have shorter wings and smaller body size in northern Australia, but no latitudinal differences in the size and shape of resident shorebirds. (6) 'Foraging ecology hypothesis': if latitudinal differences are driven by larger, or more deeply buried prey in tropical environments, we predict northern Australian shorebird populations to have longer bills than southern conspecifics. For all hypotheses, we expect weak or no differences in the size and shape of northern and southern nomadic and partially migratory species, because they move to areas with diverse climates, breed in a range of locations, travel inconsistent distances, and can relocate to new areas to find food or avoid predators.

| | | Bill length | | | Wing length | | | Body mass | | |
|---|---|---|---|---|---|---|---|---|---|---|
| | | M | N | R | M | N | R | M | N | R |
| 1 | Thermoregulatory hypothesis | ↑ | ↑ | ↑ | ↓ | ↓ | ↓ | ↓ | ↓ | ↓ |
| 2 | Developmental plasticity hypothesis | - | ↑ | ↑ | - | ↓ | ↓ | - | ↓ | ↓ |
| 3 | Starvation risk hypothesis | - | - | - | ↓ | - | ↓ | ↓ | - | ↓ |
| 4 | Predation risk hypothesis | - | - | - | ↓ | ↓ | ↓ | ↓ | ↓ | ↓ |
| 5 | Differential migration hypothesis | - | - | - | ↓ | - | - | ↓ | - | - |
| 6 | Foraging ecology hypothesis | ↑ | ↑ | ↑ | - | - | - | - | - | - |

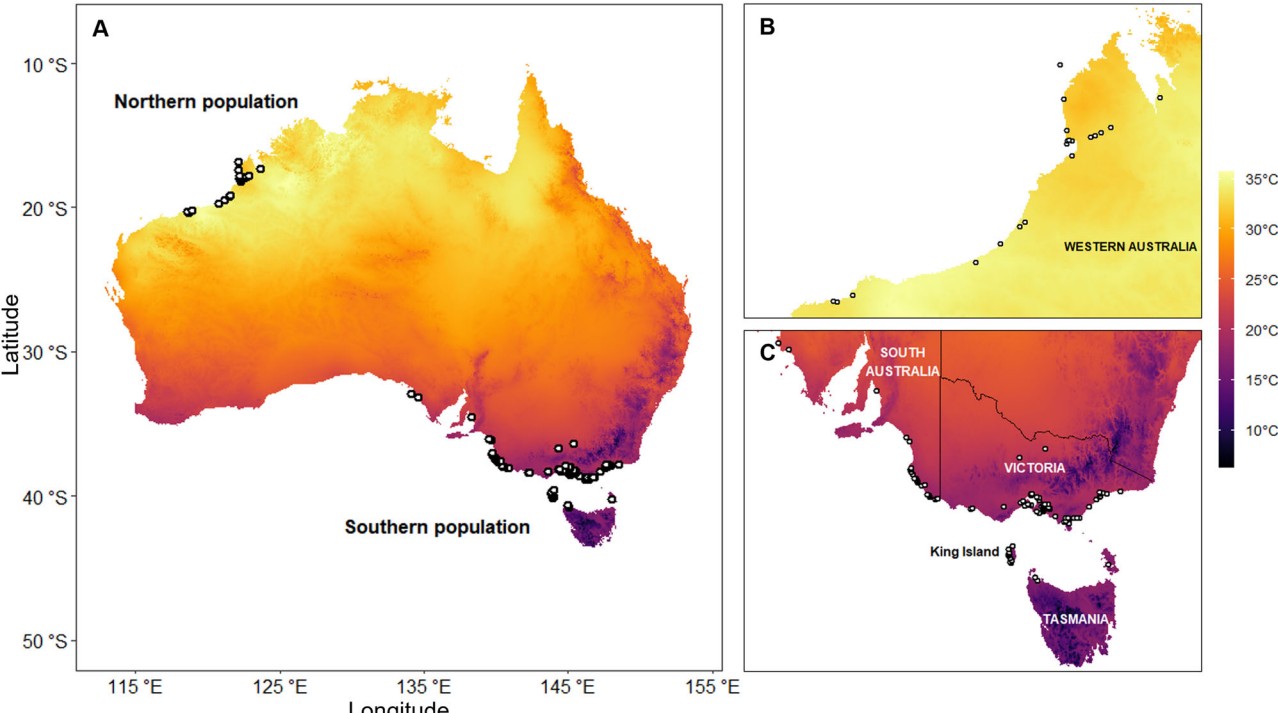

**Fig. 1 | Sampling locations for Australian shorebirds.** Field sites for shorebird research conducted by the VWSG and AWSG in **A** northern and southern Australia. Colour scale shows annual average daily maximum temperatures. **B** Northern populations were studied in hot, tropical coastal sites near Broome while **C** southern populations were studied in temperate coastal areas and wetlands along the south-eastern mainland, northern Tasmania and King Island. Climate data are from the Australian Bureau of Meteorology.

significantly lower mass for 16/18 migrant, 4/7 nomadic or partially migrant, and 4/5 resident species (Fig. 2C). None of the 30 species has significantly greater mass at the northern sites (Fig. 2C). As with bill size, the difference in wing length and mass between northern and southern populations is greatest for residents (Fig. 3C, D). See Supplementary Tables S2–S6 for full comparative statistics.

## Discussion

Consistent with Bergmann's and Allen's rules, shorebirds in tropical northern Australia are typically smaller (shorter wings, lighter body mass), and have longer bills than conspecifics in temperate southern Australia. This pattern is evident across five ecologically diverse families, including migratory species that spend early life in the Arctic or the cold deserts of Central Asia, and non-migratory species. It is likely that many factors influence latitudinal variation in animal size and shape. However, when considered in their entirety, our results best support the thermoregulatory hypothesis for Bergmann's and Allen's rules, and further suggest that genetic adaptation for thermoregulation in warm climates impacts shorebird size and shape (Box 2 and Fig. 2).

Our results are best explained by the thermoregulatory hypothesis, because northern Australian birds have both smaller bodies and longer bills than their southern Australian conspecifics (Box 2 and Fig. 2). Differences in body size and bill length are more pronounced in resident species, which experience local climate conditions year-round (Fig. 3B–D). We interpret these patterns in terms of selection to aid heat loss in tropical northern Australia: the difference is unlikely to be fully explained by adaptation to conserve heat in southern Australia because the pattern exists for both migrant and resident species, and both northern and southern Australian migrants travel to the cold northern hemisphere to breed[58]. Researchers have proposed that the selection on animal size and shape should be relaxed in the tropics, and migratory bird research emphasises morphological adaptations

for heat conservation in the northern hemisphere[8,20]. However, non-morphological adaptations could allow northern Australian migratory shorebirds to evolve smaller bodies and longer bills while conserving heat at their breeding grounds. For example, migratory shorebirds moult into thick, insulating body plumage in time for breeding[59], and could minimise heat loss by restricting blood flow to the bill surface[60]. By comparison, heat stress at the tropical non-breeding grounds is difficult to escape, as foraging times are dictated by the tide and beaches provide little protection from the sun. Migratory shorebirds in particular face pressure to build fat reserves ahead of migration, and show signs of heat stress while foraging on northern Australian coasts[61]. Northern Australian shorebirds also favour daytime roosts with relatively cool microclimates despite increased disturbance from predators[62]. While adaptation to conserve heat leads to patterns consistent with Bergmann's and Allen's rules, our results suggest that adaptations to dissipate heat further drive the evolution of animal size and shape. This interpretation fits with the suggestion that the selective climatic drivers of morphological adaptation are based on the season of highest thermal stress[63]. We speculate that adaptive evolution for thermoregulation in different climates could contribute to speciation, as some Australian shorebird species have distinct subspecies in northern and southern Australia—notably bar-tailed godwits and masked lapwings—which follow Bergmann's and Allen's rules (Fig. 2). Northern Australian masked lapwings (*Vanellus miles miles*) and sooty oystercatchers (*Haematopus fuliginosus opthalmicus*) also differ from southern subspecies by having larger areas of unfeathered skin on their faces which may be adaptations to increase heat loss[64,65].

Latitudinal patterns in the size and shape of shorebirds suggest Bergmann's and Allen's rules are driven by population-level genetic adaptation to different thermal environments. Alternatively, such patterns may derive from plastic responses to temperature during development. While this might explain differences between populations of resident species, we do not expect differences in body size and

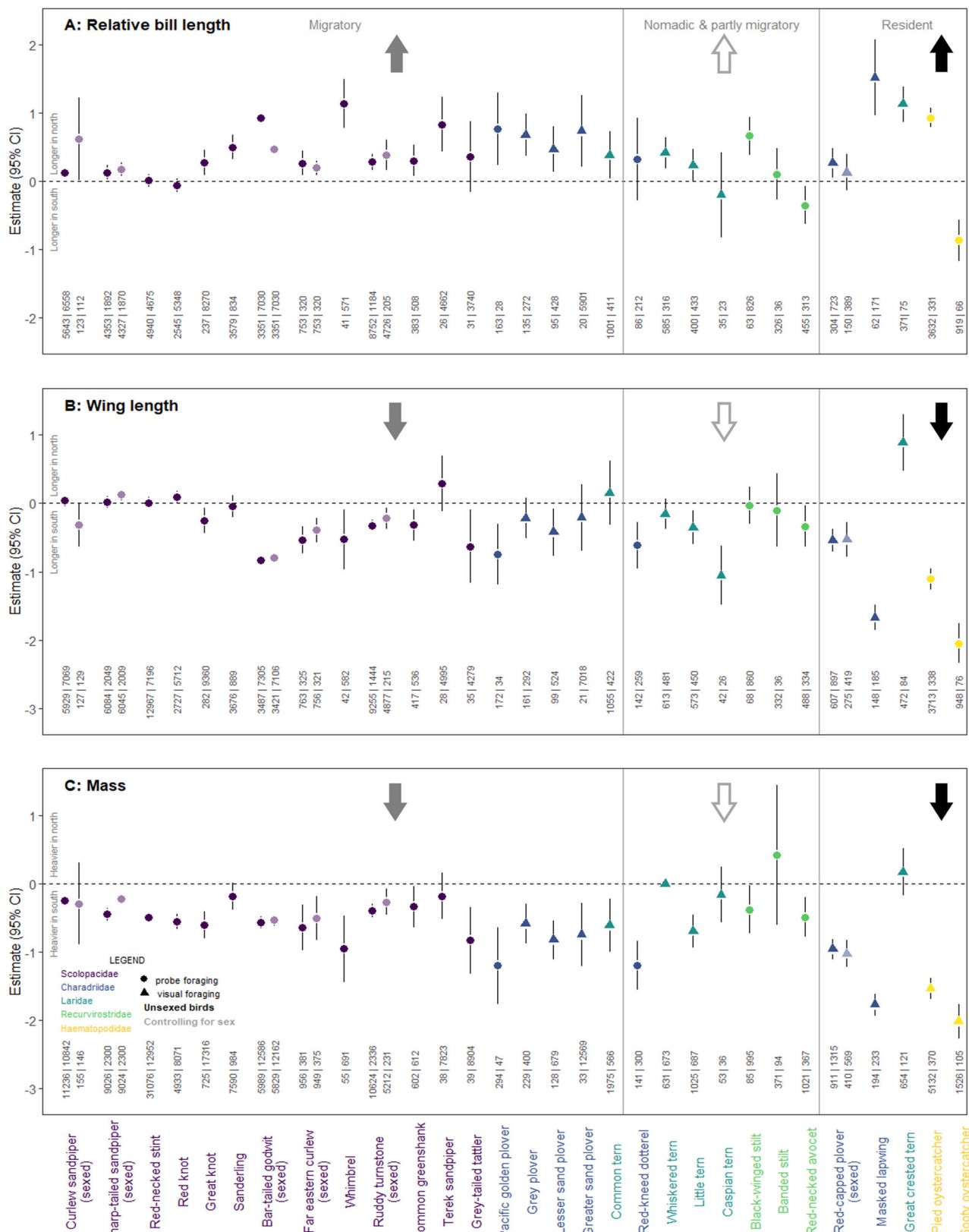

bill length in migratory species under the developmental plasticity hypothesis (Box 2). Longer bills in northern Australian migrants are unlikely to be explained by plastic responses to temperature during growth and development because Arctic-breeding migratory shorebirds appear to have fully grown bills prior to their first arrival in Australia[65,66]. Juvenile migrants are often lighter and have shorter wings

than adults during their first non-breeding season, as adult wing length is gained after their second pre-basic moult following arrival in Australia[65,66]. If warmer temperatures in northern Australia lead to reduced feather or skeletal growth, as well as reduced growth in body mass, plastic responses could explain Bergmann's rule effects for migratory and non-migratory shorebirds. However, while mass is

**Fig. 2 | Morphology of northern populations relative to southern populations of 30 shorebird species.** Effect of the northern population is shown for relative bill length (**A**), wing length (**B**) and body mass (**C**); points above the dotted line indicate larger measurements in northern populations. Bill, wing and log-transformed mass measurements were scaled and centred, so effect sizes are comparable across species. Points represent the estimates for the effect of location derived from linear mixed models. Error bars show 95% confidence intervals. Effects are shown for species with different migration behaviours (migratory, nomadic/partially migratory, and resident) and foraging methods (probing and visual). Estimates controlling for sex differences in morphology are shown for sexually dimorphic or dichromatic species. Sample sizes are shown for birds caught at each location (southern observations | northern observations). Arrows indicate predictions based on the thermoregulation hypothesis (shown in Box 2). Average effects for migratory, nomadic and partly migratory, and resident species are: 0.45, 0.17 and 0.59 for bill length; −0.25, −0.38 and −0.90 for wing length; and −0.58, −0.36 and −1.22 for mass. See Supplementary Figs. S3–S5 for boxplots showing raw data.

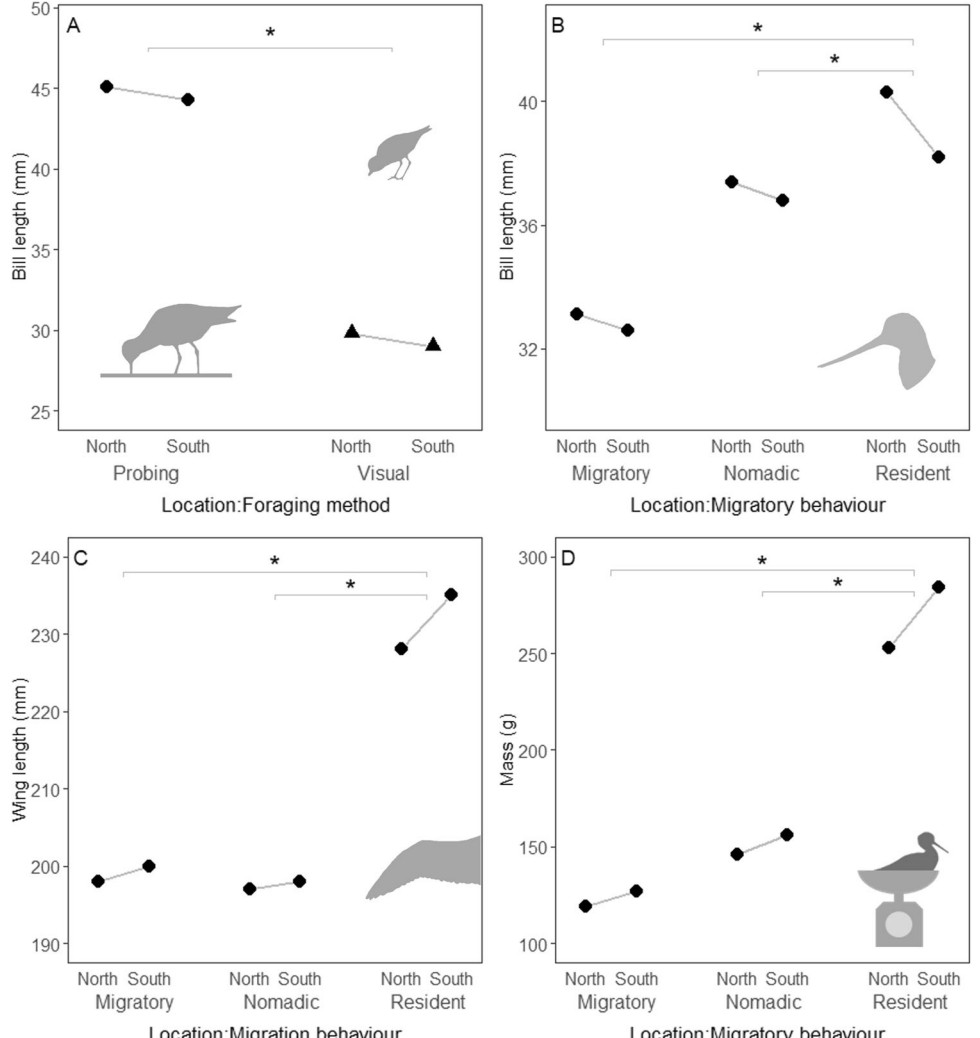

**Fig. 3 | Summary of morphological differences between different bird groups.** Model predicted means of bill length (**A**, **B**), wing length (**C**) and mass (**D**) according to population location (North = northern Australia, South = southern Australia) and (**A**) foraging method or (**B**–**D**) migratory behaviour. Predicted means are derived from phylogenetic generalised linear mixed models. Asterisks indicate comparisons where the confidence intervals do not overlap (differences between slopes).

readily adjusted according to environmental conditions, plastic responses impacting bill and limb length are thought to be confined to critical periods during the early stages of development[32,35,67]. It is possible that southern Australian populations migrate further north to colder latitudes for breeding than northern Australian populations, so that latitudinal differences in morphology observed in Australia are driven by differences in conditions at the northern hemisphere breeding grounds. To our knowledge, no evidence for such a pattern exists and indeed the only studies we know of that show spatial segregation of northern and southern Australian shorebirds at the breeding grounds show the opposite pattern[68]: far eastern curlews from southern Australia spend their breeding season in the northern hemisphere on average 4° further south of northern Australian

conspecifics[68], suggesting while differences in bill length and body size follow Bergmann and Allen's rule in Australia, the opposite pattern is shown at the breeding grounds. A similar pattern occurs in red knots[69], although they do not exhibit any latitudinal differences in wing length and bill size. Based on current knowledge, we think latitudinal differences between migratory shorebirds in Australia are unlikely to be driven by conditions at the breeding grounds because northern and southern Australian shorebird populations breed within narrow latitudinal ranges[57,66,68–70].

Temperate, southern Australian shorebirds are larger and carry greater body stores than northern conspecifics (Fig. 2B, C and Supplementary Fig. S2). Reduced day length in winter at high latitudes could select for greater body stores in seasonal environments to offset

starvation risk due to reduced foraging time[40]. However, this is unlikely to explain Bergmann's rule in Australian shorebirds because many species that follow Bergmann's rule can forage at night (e.g., species from *Charadrius*, Scolopacidae, Recurvirostridae and Haematopodidae; Fig. 2B, C). It is also possible that starvation risk is greater in tropical northern Australia than in temperate southern Australia. Low primary productivity in north Australia is proposed to explain why many migrants fly further to the southern Australian non-breeding grounds, where better foraging conditions enable faster re-fuelling[57,71,72]. Moreover, the migratory shorebird non-breeding season coincides with spring and summer in southern Australia, with only a small proportion of migrants remaining at the non-breeding grounds during Austral winter[73]. Rather than increasing body stores to compensate for increased starvation risk, low mass and body stores in northern Australian shorebirds could reflect adverse effects of reduced food availability, poor nutrition, or a more stressful environment[38,74,75], although such effects should be offset by greater competition in southern Australia[40]. Moreover, migratory shorebirds gain substantial mass ahead migration, indicating northern Australian shorebirds are capable of increasing energy stores[76]. The winter starvation risk hypothesis is perhaps most relevant for explaining body size variation for diurnal foraging species in temperate regions with cold winters, especially where foraging is restricted by snow and ice[40]; we propose that further research is needed to investigate geographical variation in starvation risk at a global scale.

High predation risk could explain why northern Australian shorebirds have lower body stores and shorter wings than their southern conspecifics, as these traits enhance agility and escape capacity[46,77,78]. Predation risk is generally thought to be higher at lower latitudes due to a greater abundance of raptors and increased trophic interactions in the tropics[45,79] but see[80]. If predation risk drives latitudinal patterns observed here, we would expect latitudinal differences in body stores and wing length to be greatest for small, ground-foraging species, as they are generally more vulnerable to avian predators[81–83]. We find that small, ground-foraging species such as red-necked stints and red-capped plovers indeed follow Bergmann's rule. Birds of this size are targeted by a wide range of predators, including smaller birds of prey, such as nankeen kestrels (*Falco cenchroides*) and Australian hobbies (*F. longipennis*)[65]. However, we also see strong Bergmann's rule effects for shorebirds that are too large to be hunted by smaller predators, including sooty and pied oystercatchers, and birds that forage in flight, such as little terns, despite the expectation they are more difficult for predators to catch[84] (Fig. 2B, C). Masked lapwings show the greatest effect of latitude on body mass and wing length but are likely among the least vulnerable species to predators, as they are highly aggressive and possess sharp, defensive wing spurs[65]. While the starvation and predation risk hypotheses could explain latitudinal patterns in shorebird body size, an additional explanation is needed for the accompanying latitudinal differences in bill size.

The differential migration hypothesis cannot fully explain Bergmann's rule in Australian shorebirds because we find patterns consistent with Bergmann's rule in non-migratory shorebirds (Box 2 and Fig. 2). This suggests longer wings and larger bodies in southern Australia are not entirely due to selection for increased flight efficiency on their journey to the northern hemisphere, or morphological sorting of different sized migrants at the non-breeding grounds. We also find patterns consistent with Bergmann's rule in migratory species while controlling for differences in body size according to age and sex, and among sexually monomorphic migratory species. Therefore, patterns among migratory species are not fully explained by competition for preferred non-breeding grounds between larger adults and smaller juveniles[85] or sex differences in migration behaviour in sexually dimorphic species[50]. Nevertheless, such effects could further contribute to latitudinal differences in body size. For example, female far

eastern curlews show a stronger preference for the southern Australian non-breeding grounds than smaller male conspecifics[58].

While diet and foraging behaviours clearly have a profound effect on the evolution of bill size and shape on a global scale[86], our results suggest latitudinal trends in bill length may be further influenced by thermoregulatory adaptation. We found consistent evidence for Allen's rule across species with different diets and foraging behaviour (Fig. 2A). Both probing and visual foragers show similar increases in bill sizes in northern populations (Fig. 3A). Longer bills occur in northern populations of species that dive for fish (e.g., common terns), peck invertebrates from surfaces (e.g., red-capped plover) and probe for prey beneath sand and water (e.g., whimbrel, black-winged stilt). Past studies further demonstrate longer bills in warmer regions in nectar-feeding honeyeaters[25], vertebrate-hunting raptors[23], frugivorous toucans[9] and granivorous parrots[87]. Evidence for Allen's rule across species with different foraging methods and diets—and corresponding patterns in body size consistent with Bergmann's rule—best supports the single explanation provided by the thermoregulatory hypothesis. Future research could explore whether foraging specialisation influences the capacity of bills to be modified for thermoregulation in different climates.

We propose that evidence for Bergmann's rule and Allen's rule across ecologically diverse species, with different migration behaviour and foraging strategies, favours the single, overarching explanation supplied by the thermoregulatory hypothesis. While patterns we observe could be accounted for through multiple alternative explanations acting in combination (e.g., developmental plasticity, starvation risk, predation risk, differential migration, foraging ecology), the thermoregulatory hypothesis provides the most parsimonious explanation for the broad consistency of Bergmann's and Allen's rules across Australian shorebirds.

## Methods

We tested evidence for Bergmann's and Allen's rules by comparing bill length and body size of conspecific shorebirds in northern and southern Australia. Comparison of shorebirds in two distinct regions allows us to test whether latitudinal effects differ across ecologically diverse species (Fig. 1). When testing Bergmann's Rule, we estimated body size using wing length and body mass. Wing length and body mass are strongly correlated ($r = 0.82$, $N = 118,725$ across all species). Wing length is a good indicator of body size[88], is less prone to seasonal variation than body mass[89] and is widely used in past research, including Bergmann's original observation of latitudinal trends[4]. Meanwhile, body mass directly impacts the surface area to mass ratio, which is relevant to a thermal explanation for Bergmann's rule[17] since most heat exchange from birds occurs via their body[90]. Latitudinal differences in wing length can also be explained by other factors, independent of wing length being a proxy for body size but relevant to the hypotheses outlined in Box 2, such as predation risk[47] or migration distance[91]. When testing Allen's rule, we assessed whether northern Australian birds have longer bills relative to their body size ('relative bill length', controlling for variation in body size by including wing length as a covariate in models) and longer bills regardless of variation in body size ('absolute bill length'). Assessing relative bill length allows us to determine whether northern birds have longer bills relative to their bodies to promote heat loss. However, changes in relative bill length may be the result of selection for smaller body size, rather than larger bill size, making it difficult to disentangle responses under our hypothetical framework; assessing absolute bill length allows us to test whether bills are longer independent of drivers that may influence body size.

### Study site and species

Bill length, wing length and body mass data were collected as part of ongoing banding programs by the VWSG and AWSG[92,93]. These

groups have been catching and banding birds with uniquely numbered leg bands—allowing identification of individuals—and collecting morphometric data since 1975. Shorebirds are regularly sampled in coastal regions of north-western Australia, close to Broome ('northern' sites) and south-eastern Australia, in Victoria, South Australia, King Island and northern Tasmania ('southern' sites; Fig. 1). Northern sites are regularly exposed to high temperatures (Broome mean maximum temperature = 32.2 °C annual; 34.3 °C warmest month), while southern sites are temperate, with mild winters and warm summers with occasional hot weather (Mornington, Victoria mean maximum temperature = 18.9 °C annual; 25.0 °C warmest month; King Island, mean maximum temperature = 17.0 °C annual; 21.2 °C warmest month). Individual shorebirds typically migrate to the same region of Australia for the non-breeding season; according to the shorebird dataset used here, 139 of 43,560 (0.3%) migratory individuals recaptured across different years were caught in both northern and southern Australia, possibly due to a change in non-breeding grounds across different years, mistakes when recording band numbers, or because of a stopover in northern Australia on migration from or to southern Australia. Individuals recaptured in both northern and southern Australia across different years include bar-tailed godwits, curlew sandpipers, great knots, red-necked stints, red knots, ruddy turnstones and sanderlings.

We selected species for analysis if they were sampled in both northern and southern Australia on at least 20 occasions. Our analyses include data from 30 species, comprising 13 sandpipers (Scolopacidae), seven plovers and lapwings (Charadriidae), five terns (Laridae), three stilts and avocets (Recurvirostridae) and two oystercatchers (Haematopodidae) (Supplementary Table S1). Across the dataset, birds have been captured during all months of the year and the same individuals have been sampled 1–15 times (mean number of captures = 1.5, SD = 0.9). Sampling effort has been similar across years in northern and southern Australia (Supplementary Fig. S1).

### Field methods

Members of the VWSG and AWSG typically caught shorebirds using cannon-nets while the birds were roosting during high tide. Bill length was measured as the exposed culmen (tip of bill to the base of feathers) to the nearest 0.1 mm using callipers. Wing (maximum chord) length was measured while straightened and flattened using a butt-ended ruler, from the shoulder to the tip of the longest primary feather, to the nearest 1 mm (larger species) or 0.1 mm (smaller species). Wing length measurements were excluded for birds moulting their ninth or tenth primary wing feather. Birds were weighed using scales to determine body mass to the nearest 1 g (larger species) or 0.1 g (smaller species). Morphometric data were cleaned using standard procedures before analyses and blind to the birds' location. Repeatability of measurements based on recapture data is high and measurement differences in northern and southern Australia are extremely unlikely to explain the patterns reported in our results due to the size of the dataset and the close affiliation between the VWSG and AWSG (the AWSG was founded by the former), meaning many of the same people were responsible for collecting morphological data in both regions (see Supplementary Note 1 for additional details on data processing and repeatability).

Sex was estimated in the field for a subset of individuals from six species, according to sex differences in plumage characteristics (red-capped plover) or a combination of plumage characteristics and morphology (bar-tailed godwit, curlew sandpiper, far eastern curlew, ruddy turnstone and sharp-tailed sandpiper). We later assigned sexes to unsexed individuals of sexually dimorphic species according to differences in bill size (bar-tailed godwit, far eastern curlew) and body mass (sharp-tailed sandpiper), following information in the Handbook

of Australian, New Zealand and Antarctic Birds ('HANZAB')[65,66] and patterns in our data (Supplementary Note 1).

Age (first-year juveniles vs. adults ≥2 years) was determined by feather wear[94]. Age was not assigned for all individuals on every capture event, so we also estimated age based on repeated captures of the same individual. We assumed birds that were not aged in the field and that could not be aged by recaptures were adults (1%, 1% and 2% of samples for bill, wing length and mass analyses, respectively). Young birds with downy feathers were excluded from all analyses.

### Ethics compliance

All research protocols over the 46 years of data collection were approved by animal ethics committees registered with the Australian states in which fieldwork was conducted, including the Department of Environment, Zoos SA, and South Australian Museum combined (South Australia); Department of Primary Industries, Parks, Water and Environment (Tasmania); Philip Island Nature Parks (Victoria); Department of Primary Industries and Regional Development WA, Division of Agriculture and Food (Western Australia). Other licences, permits and approvals as required under Australian state and commonwealth legislation were also obtained from appropriate bodies including Department of Environment and Water (South Australia); Department of Primary Industries, Park Water and Environment (Tasmania); Department of Environment, Land, Water and Planning, and Parks Victoria (Victoria); Department of Biodiversity Conservation and Attractions (Western Australia). All activities are registered with the Australian Bird and Bat Banding Scheme that provides metal bands for the projects.

### Migration and foraging data

We collected data on migration and foraging behaviour using the information in the HANZAB. We classed species as (i) 'resident' if they were described as 'resident,' 'mostly resident', 'sedentary' or 'mostly sedentary' or do not regularly undergo long-distance movements. These species may form roaming winter flocks, but a large part of the population typically remains in the same general area. We considered species to be (ii) 'nomadic' if they frequently display 'nomadic', 'dispersive' or 'opportunistic,' movements, or fly long distances in response to rainfall. We considered species to be (iii) 'partly migratory' if described as 'partly migratory' or 'mostly migratory,' or have a combination of migratory and non-migratory Australian populations. We combined nomadic and partly migratory species in the same category for our comparative analyses (see below) because we predict both groups to show similar latitudinal differences in morphology based on the hypotheses in Box 2. Finally, we classed species as (iv) 'migratory' if they undergo regular, seasonal movements between separate non-breeding and breeding grounds. These species fly long distances to breed in the Arctic or Northern and Central Asia.

While geographic patterns in prey size and behaviour could select for longer bills in northern Australia, the foraging ecology hypothesis is less strongly supported if similar latitudinal patterns in bill length are detected across species with diverse foraging behaviours. We therefore scored foraging behaviour per species as either visual searching for prey ('visual foraging') or a combination of visual hunting and tactile probing for prey beneath surfaces ('probing foraging'). We considered species to be visual foragers if they forage using at least one of the behaviours: 'peck', 'pick', 'tap', 'jab', 'glean', 'plover-style', 'prise', 'hawk', 'dip' and 'dive' but seldom use probing behaviours (see below). Visual foraging species include many of the plovers, which typically peck prey off surfaces, and terns, which catch invertebrates in flight and dive for fish. Probing species were described as using at least one of the following foraging behaviours: 'probe', 'scythe', 'sandpiper-style', 'stitch', 'sew', 'plough' and 'sweep;' all probing species employ a combination of visual and tactile techniques. Probing foragers include the sandpipers, oystercatchers, stilts and avocets, which use their bills

to sense prey hidden beneath sand, mud, or water. There was little information on the foraging behaviour of lesser sand plovers, but they appear to use similar methods as congeneric greater sand plovers (visual foraging[95]).

## Analysis

We analysed differences in the size and shape of shorebirds in northern and southern Australia using both within-species comparisons, and across all 30 species using a phylogenetic comparative approach. Analyses were conducted using R (version 4.0.2).

**Within-species analyses.** For the within-species comparisons, we compared the size and shape of northern and southern shorebird populations using separate linear mixed models for three response variables (bill length, wing length and body mass), for each of the 30 species. Models were run using the R package 'lme4'[96] version 1.1–26. We log-10 transformed body mass to improve normality and centred and scaled the response variables 'bill length', 'wing length' and 'log-10 mass' to facilitate comparison of effect sizes across species. In addition to our key variable of interest—location where birds were sampled (northern vs. southern sites)—we included 'age' (adult vs. juvenile), 'season' (wet/spring-summer vs. dry/autumn-winter) and 'sample year' as fixed effects to control for age differences in body size and shape, temporal variation in climate, and seasonal changes in body size and shape (e.g., according to seasonal fattening, feather and bill wear). For bill length analyses, we included wing length as a fixed effect to assess differences in bill length relative to body size. For species with sexed individuals, we ran additional models including sex as a fixed effect to account for sex differences in morphology (excluding samples from unsexed individuals). To control for repeated samples within the same individuals, we included individual ID 'band number,' as a random effect. Band number was excluded from the analyses when repeated captures made up less than 3% of the total sample; excluding samples from repeated captures from these analyses does not qualitatively impact the results. Sample year was excluded from models assessing differences in morphology for red-kneed dotterels and banded stilts due to strong correlations between year and location ($r \geq 0.8$). Data were originally recorded on datasheets by the same team of people measuring birds for the same capture event on a given day. We therefore included 'datasheet ID' as a random effect to control for variation in measuring technique, and possible sampling effects associated with a particular catch on a given date.

**Across species analyses.** We tested for overall differences in size and shape of northern and southern shorebirds across 30 species using Bayesian phylogenetic generalised linear mixed models. Bayesian models were run using the R packages 'INLA'[97] version 21.02.23 and 'phyr'[98] version 1.1.0 with a complexity penalising prior described by Simpson et al.[99] and implemented as the 'automatic prior' in INLA and phyr[98]. Separate models were run for response variables bill length (mm), wing length (mm) and body mass (g). All three response variables were log-10 transformed to improve normality and homogeneity of variance. We present back-transformed model predicted means in the results. Models included the fixed effects 'sample location', 'age', 'season' and 'sample year' as described above, as well as 'migration behaviour' (migratory, nomadic/partially migratory, and resident). We included 'wing length' (as above) and 'foraging method' (visual foraging vs. probing) for bill length models. We included random effects for 'species ID' to account for repeated samples collected within the same species, and phylogeny, to account for species relatedness. To account for phylogenetic uncertainty, we built a 'maximum credible tree' using a sample of 1000 trees from Jetz et al.[100] and the 'phangorn' R package[101] version 2.7.0. To assess differences in absolute bill length, we repeated bill length models excluding wing length as a covariate.

Differences in body stores are relevant to the starvation and predation risk hypotheses, where northern birds are predicted to reduce body stores due to lower starvation risk and increased predation risk[40]. We therefore repeated our model assessing body mass (described above) but included wing length as a fixed effect (i.e., a proxy for structural body size), to estimate latitudinal variation in body stores.

We ran additional models including the same effects described above, and the interactions between sample location (north vs. south) and migration behaviour, to assess whether latitudinal differences in bill length and body size varied with migration behaviour. We also analysed variation in bill length according to the interaction between sample location and foraging method to determine whether different foraging methods influence latitudinal patterns in bill length. Interactions were tested in separate models to ensure model convergence.

## Reporting summary

Further information on research design is available in the Nature Research Reporting Summary linked to this article.

## Data availability

The data used in the analyses for this study are available through Dryad data depository (https://doi.org/10.5061/dryad.xsj3tx9j5)[102].

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

## Acknowledgements

This research is possible thanks to the extraordinary efforts and expertise of community scientists belonging to the Victorian Wader Study Group and the Australasian Wader Studies Group. We thank the members for allowing us access to their dataset, for making us welcome on field trips and for their valuable feedback on the manuscript. A list of the members who have contributed to this dataset over 46 years can be found in Supplementary Note 2. For the work in northern Western Australia, we acknowledge the Yawuru People via the offices of Nyamba Buru Yawuru Limited for permission to catch birds on the shores of Roebuck Bay, traditional lands of the Yawuru people. We also acknowledge the Karajarri and Nyangumarta people for permission to catch and mark birds on the shores of 80 Mile Beach, traditional lands of the Karajarri and Nyangumarta people. For the work in southern Australia, we acknowledge the Gunaikurnai, Wadawarrung, Bunurong, Eastern Maar, Gunditjmara and other first nations peoples for allowing us to work on their traditional lands. Thanks to three anonymous reviewers, Danny Rogers, Sara Ryding and Ryan Barnaby for helpful feedback and discussion, to Daijiang Li and Håvard Rue for advice on using 'phyr' and 'INLA' in R, and to Robert Moore and Jake Tyers for IT support. This research was supported by an Australian Research Council Discovery Project grant (DP190101244) to M.R.E.S., M.K. and G.J.T. C.J.H. was funded by the Spinoza Premium of Netherlands Organisation Prize for Scientific Research awarded to Theunis Piersma (2014–2017), WWF Netherlands (2010–2021) and MAVA, Foundation pour la nature (2018). This paper is dedicated to the late Clive Minton (1934–2019), founder of the Victorian Wader Study Group and the Australasian Wader Studies Group, and the driving force behind the banding studies that collected the data we report on here.

## Author contributions

A.M., G.J.T., M.K. and M.R.E.S. conceived the study; members of the Australasian Wader Studies Group (AWSG) and Victorian Wader Study Group (VWSG) collected the field data over 46 years, including C.J.H., M.C., M.K., R.A. and R.J.; A.M. collected data from handbooks; A.M. and M.R.E.S. analysed the data; A.M. wrote the manuscript, with contributions from G.J.T., M.K. and M.R.E.S.

## Competing interests

The authors declare no competing interests.

## Additional information

## Victorian Wader Study Group

Alexandra McQueen [1], Marcel Klaassen [2], Robyn Atkinson[4], Roz Jessop[4] & Maureen Christie[6]

## Australasian Wader Studies Group

Marcel Klaassen [2], Robyn Atkinson[4], Roz Jessop[4], Chris J. Hassell[5] & Maureen Christie[6]

Full lists of members and their affiliations appear in the Supplementary Information.

