## [Peer Review File · Nature Communications]

Thermal adaptation best explains Bergmann's and Allen's Rules across ecologically diverse shorebirdsReviewers' Comments:

Reviewer #1:

Remarks to the Author:

In their manuscript, "Thermal adaptation best explains Bergmann's and Allen's Rules across ecologically-diverse shorebirds", McQueen et al. assess the drivers of size and appendage length in an impressively diverse and long-term dataset of shorebirds. They find that bill length and body size conform phenomenologically to the expectations derived from the Rules, and attribute this to adaptation to warm climates. The authors have written a wonderful paper – it's a fascinating topic, an extremely clever/creative use of this natural experiment and amazing dataset, and I think it is destined to be of broad interest to researchers in the field. In particular, I think the framework of trying to evaluate the various competing hypotheses is incredibly effective in highlighting the various potential drivers of the observed changes. That being said, the clarity that came with the framework raised some – what I would consider to be fairly major – questions. I would not be the least bit surprised to learn that these are born out of my lack of familiarity with the system (in which case, I hope the authors will feel free to say so), but I think they necessitate further consideration (or simply expanded justification in the text for those of us who are not familiar with the system), as they do seem to me to speak to the fundamentals of the interpretation of the paper.

1) The biggest question I have is regarding the potential complications of not knowing the variation in temperatures on the breeding grounds for migratory species. While the authors state that they do not think this is possible, given the constrained latitudinal range of the breeding grounds, I am not convinced. For example, in the references provided for godwits, at a broad scale this seems true (one paper summarizes that they breed in North Eastern Russia and Alaska), however the detailed tracking data tell a much more nuanced story that I think complicates the findings of the paper. *L. lapponica menzbieri* is in the northern population, with individuals that likely breed in the Yellow Sea region, while *L. l. baueri* is in the southern population, with individuals that likely breed in Western Alaska. These places are not really similar latitudinally, and certainly differ in their climates, with the Yellow Sea region considerably warmer than the North Slope of Alaska. As a result, it seems to me that one possibility is that developmental plasticity is driving these patterns. If, for example, migratory species are breeding in relatively similar environments (compared to the differences between north and south Australia), but Northern Australian populations breed in localities that are slightly warmer than migratory species from Southern Australia – as is likely the case with *L. lapponica* – wouldn't you expect to see exactly the pattern you find? Smaller longer-winged birds in Northern Australia, with the pattern slightly more pronounced for resident species? In other words, wouldn't the predictions in Fig 1 be identical for hypotheses 1 and 2?

2) I'm not sure that I completely understand the rationale for assuming wing length and body mass will change in lock step. While wing length may often be used as an index for body size, and I agree that there are limitations associated with mass, there is also recent work in North America demonstrating that temperature drives reductions in body size, but not wing length. Thus, it seems at least possible – if not likely – that wing length and body size might have contrasting relationships with temperature. At the very least some consideration of this seems necessary.

3) I'm not entirely sure I understand how some of the categories in Fig 1 relate to each other. For example, I assume that relative bill length is bill length relative to mass? If so, couldn't reduced body size drive increases in relative bill length? So for example, the predation risk hypothesis would drive reductions in mass and as a result if bill is constrained by foraging behavior, relative bill length would be expected to increase across all of the categories of migration? This seems similar for the starvation risk hypothesis; nomadic species are not predicted to change in size under this hypothesis, but it is not clear to me why that is. In other words, it seems that at least 3 or 4 of the hypotheses may actually generate the same set of predictions?

4) I'm also not sure that body mass is necessarily a good estimate of the surface area to mass ratio

that matters for birds; given that essentially all heat exchange occurs from the appendages, isn't the ratio of wing/bill length to mass a more meaningful surface area:mass ratio?

Again, I greatly enjoyed reading this manuscript, and I think this is likely to be a broadly important study that is building on a long history of amazing work to come out of this group.

Reviewer #2:

Remarks to the Author:

This is the first time in my career that I have ever seen a presumed first-submission manuscript and thought it ready for acceptance. Congratulations to the authors – this is a lovely piece of science.

This manuscript considers two ecogeographic rules (Bergmann's and Allen's rules, that species at higher latitudes will be, respectively, larger-bodied and smaller-limbed) in 30 species of shorebirds using a stagger 20,000 datapoints across 46 years. The authors consider six possible evolutionary drivers of these patterns, and determine of those that selection for thermoregulation best explains the morphological differences observed in this dataset. These six hypotheses are very clearly explained and well-grounded in the literature; the models used to demonstrate the statistical relationships are appropriate and well-documented. Ecogeographical rules seem to be making something of a comeback recently, and this is a very elegant investigation of a well-selected system. I could see this study featuring in a lot of textbooks.

(Also, as a small side-note, I appreciate the use of the phrase "community scientists".)

My one and only quibble is the apparent assumption that plasticity is not adaptive. This position obviously exists in the field evolutionary biology, but it is controversial, and I would worry that it would detract from the main study used here. The two places I noticed it most were L 28 and L 374-376; these could be easily rephrased if the authors wish to not wade into this particular fight. (For example, in the former, "...improved thermoregulation including plastic responses to..."; in the latter, "driven by population-level genetic adaptation" or "driven by adaptation to...In particular, such patterns"?) Alternatively, I would certainly find a clear discussion of whether or not plasticity is adaptive in light of ecogeographical rules to be fascinating, but I would recommend that the authors make it clearer in-text that this is the position they're taking.

Anyway, congratulations to the authors – I really enjoyed reading this manuscript.

Reviewer #3:

Remarks to the Author:

The present study investigates two historical issues in biogeography, Allen's and Bergmann's rules, using many species of Australian shorebirds as study system and tested patterns of variation in bill and body size at both within-species and cross-species levels. The authors did an unprecedented effort to collect a huge number of individuals of 30 species (the total sample size is >200'000 individuals measured) and compared bill length, body size and body mass between populations located in hot, tropical northern Australia vs. those located in temperate, southern Australia. Most of the analysed species showed pattern of variation compatible with both ecogeographical rules, with populations in the north having longer bills and smaller bodies than conspecifics from the south. The authors also tested for multiple competing hypotheses about the evolution of bill and body size and concluded that, given that similar patterns are observed in species showing different migration and foraging strategies, the most parsimonious mechanism explaining the overall phenotypic variation is thermoregulation (i.e. smaller bills and larger bodies increase insulation in colder climates, and vice versa). The study is interesting, the manuscript is well written, the predictions are clearly stated, and

the results are clear. Also considering the large taxonomic and geographic coverage of the species included in the analyses, I think that the study has merits that makes it publishable. However, there are some issues that I would like to raise below in order to improve the quality and the readability of the paper.

1) My main comment concerns the interpretation of the results. Indeed, the study unequivocally shows that Allen's and Bergmann's rules apply to this various group of birds, irrespective of the ecological features of each species. The fact that species showing different migration and feeding strategies show coherent pattern of geographic variation in phenotypic traits is very important because it shows the generality of these patterns. However, I would be more careful in the interpretation of the mechanism(s) behind the observed variation. Indeed, all the competing hypotheses proposed to explain the spatial variation in bill and body size have predictions in the same direction (i.e. increase in body size and decrease in bill length in colder climates) and most of them also predict the same (or similar) strength of the effect for the same category of birds (e.g. resident species). Given the nature of the data used, which are necessarily correlative, the study cannot test which of the competing hypotheses is best supported. On the whole, I could agree with the authors that thermoregulation seems to be the most parsimonious mechanism explaining the overall variation. However, with the present data it is not possible to exclude that different mechanisms can explain variation in bill and/or body size in some categories of birds (e.g. it is possible that phenotypic plasticity during growth explains variation in bill length and body size in resident species more than thermoregulation). Or even that multiple mechanisms act together to produce the observed patterns (e.g. phenotypic plasticity, thermoregulation and starvation risk may explain variation in body size in resident species). The authors used a quite cautious wording in the title and in the very last paragraph of the Discussion, and I agree with such an approach. However, I think that the entire Discussion should be more clear about such a possible multiple concomitant effects, and I am not sure that the use of subheading helps the understanding of the possible complex patterns of variation, with multiple selective forces potentially acting together.

2) A second main point concerns how the authors treated their original data. I think the authors should make more effort to describe their original dataset and provide some additional information that are very relevant for the reader to understand if data are reliable and eventually if the procedures used for data collection may have somehow affected their conclusions. I think that the authors should easily be able to provide all the relevant information that I briefly summarize below.

2a. As clearly stated by the authors, their database may suffer from the fact that measures were taken by multiple measurers, potentially posing problems in the reliability of the data used. Because a small fraction of the individuals have been captured multiple times, I would suggest to provide some analyses of repeatability of the measurements taken by different measurers. Even if recaptures represent a very small fraction of the individuals, considering the very large capture effort, the authors could rely on an adequate sample of individuals to perform these analyses and therefore to show that their data are solid and results are robust. In addition, I would also add that an understanding of measurement error would help to better interpret the biological relevancy of the results obtained, especially because the relative difference in the analysed traits between southern and northern populations is far from being large (< 2% for bill and wing length). It would therefore be important to know whether measurement error is within the range of the reported spatial variation, but even if it is not (hopefully), it would be important for the readers to know this.

2b. I think that some information is needed also about the consistency of wintering grounds along the life of single individuals of migratory species. In practice, do the same individuals always winter in the same area? Or is there any evidence that some individuals (at least in some species) can overwinter on the northern coast in some years and in the southern one in the other years? This is a crucial point to understand if the premises of the study are met for (all or most of) the migratory species. I think that this information can be extracted again from multiple captures of the same individuals in different

years.

2c. Another quite tricky point concern the year of capture of different individuals. Although the authors correctly included year of capture as a covariate in the models, I was wondering if the distribution of captures in the two locations is constant over time. Indeed, if thermoregulation is the main cause of geographic variation in these traits, we should expect a variation in bill size (increase), body size (decrease) and body mass (decrease) over time due to the increase in temperatures as a consequence of global warming. I note that trends compatible with this idea, although statistically non-significant (except for body mass), are presented in the results (Tables S2-5). An uneven temporal distribution of the captures may have therefore affected the results (e.g. the observed patterns should have been obtained if a larger capture effort was done in recent years in the north). Thus, some information about this should be included.

3) A final consideration concerns statistical analyses. Let me start by saying that statistics are state of the art, and therefore I do not find any major flaw about how the models have been performed and about the packages used. However, it is not clear to me why the analyses of bill variation including interaction terms (i.e. location*migration and location*foraging) were treated in separate models. In my opinion, they should be included in a single model containing multiple interaction terms. I would encourage the authors to provide a single full model or, at least, to justify why it is better to split the analyses.

Minor comments:

LL 73-76. It might be worthy to mention that a positive covariation between the size of the predators and that of their prey should also be expected. There is a large body of evidence about such an expected trend, including a few empirical studies showing within-species variation in body size according to spatial variation in the size of the prey. If the authors do not intend to add this one among the competing hypothesis on variation in body size, I think that they should at least acknowledge the reader about it.

Fig. 3. I think that the figure would benefit from the inclusion of average effect sizes resulting from the across-species analyses per migration category and one indicating the overall effect on all the species used. This would help researchers to compare biological effects across different studies.

LL 378-381. This is not necessary true, at least for some of species. The assumption that all the populations of all the species breed under similar climatic conditions can be questioned, as many species typically show a large breeding area covering a large latitudinal range. This could be a potential problem for the species showing a large migratory connectivity (the spatial and temporal linkages of individuals and populations between seasons that result from migratory movements). Do the authors have some evidence that (some of) the species included in the analyses display a large migratory connectivity with populations breeding in warmer climates also wintering in warmer areas? However, some discussion about this point is recommended.

LL 428-430. I do not fully understand this sentence. Indeed, according to Fig. 1, both the starvation and predation risk hypotheses do not have any prediction about variation in bill length. We cannot exclude that body size would vary according to either or both of these hypotheses and bill length varies according to thermoregulation. This comment is fully related to my first main comment.

Effect of year on body mass. I am surprised that this significant effect, as shown in Table S5, is not discussed at all. I think it is very important to include some interpretations of such an effect in the Discussion, also because it seems to be in line with the interpretation that thermoregulation is the

main driver of this trait.

RESPONSE TO REVIEWER COMMENTS

We thank all three reviewers for their encouraging and constructive comments. We have endeavoured to address all their comments and amend the manuscript as appropriate, as indicated below. In the few cases where we haven't adopted their suggestions, we provide further explanation below. Our responses are in blue italic. We have uploaded a revised MS with track changes and a clean copy without showing track changes; please note that line numbers refer to the track changes version of the text.

Reviewer #1 (Remarks to the Author):

In their manuscript, "Thermal adaptation best explains Bergmann's and Allen's Rules across ecologically-diverse shorebirds", McQueen et al. assess the drivers of size and appendage length in an impressively diverse and long-term dataset of shorebirds. They find that bill length and body size conform phenomenologically to the expectations derived from the Rules, and attribute this to adaptation to warm climates. The authors have written a wonderful paper – it's a fascinating topic, an extremely clever/creative use of this natural experiment and amazing dataset, and I think it is destined to be of broad interest to researchers in the field. In particular, I think the framework of trying to evaluate the various competing hypotheses is incredibly effective in highlighting the various potential drivers of the observed changes. That being said, the clarity that came with the framework raised some – what I would consider to be fairly major – questions. I would not be the least bit surprised to learn that these are born out of my lack of familiarity with the system (in which case, I hope the authors will feel free to say so), but I think they necessitate further consideration (or simply expanded justification in the text for those of us who are not familiar with the system), as they do seem to me to speak to the fundamentals of the interpretation of the paper.

We thank the reviewer for their constructive feedback.

1) The biggest question I have is regarding the potential complications of not knowing the variation in temperatures on the breeding grounds for migratory species. While the authors state that they do not think this is possible, given the constrained latitudinal range of the breeding grounds, I am not convinced. For example, in the references provided for godwits, at a broad scale this seems true (one paper summarizes that they breed in North Eastern Russia and Alaska), however the detailed tracking data tell a much more nuanced story that I think complicates the findings of the paper. *L. lapponica menzibieri* is in the northern population, with individuals that likely breed in the Yellow Sea region, while *L. l. baueri* is in the southern population, with individuals that likely breed in Western Alaska. These places are not really similar latitudinally, and certainly differ in their climates, with the Yellow Sea region considerably warmer than the North Slope of Alaska. As a result, it seems to me that one possibility is that developmental plasticity is driving these patterns. If, for example, migratory species are breeding in relatively similar environments (compared to the differences between north and south Australia), but Northern Australian populations breed in localities that are slightly warmer than migratory species from Southern Australia – as is likely the case with *L. lapponica* – wouldn't you expect to see exactly the pattern you find? Smaller longer-winged birds in Northern Australia, with the pattern slightly more pronounced for resident species? In other words, wouldn't the predictions in Fig 1 be identical for hypotheses 1 and 2?

We believe the specifics in this comment come from a misunderstanding of the complex movement ecology of bar-tailed godwits, which breed in north-eastern Russia and Alaska but travel via the Yellow Sea during migration (note they definitely do not breed in the Yellow Sea region as the

*Reviewer suggests above). However, more generally, the Reviewer raises a valid point about the possibility of migratory connectivity and possible spatial segregation at the breeding grounds of populations spending the non-breeding season in northern and southern Australia. We, however, maintain that this is unlikely to explain the patterns we observe in our results, because shorebirds that winter in Australia appear to breed within narrow latitudinal ranges (again, unlike suggested by the Reviewer above). Unfortunately, the tracking data available for shorebirds from the northern and southern Australian non-breeding grounds are scant. We do now include a reference to a 2011 study by one of the co-authors that examined these differences in red knots (Hassell et al 2011 *BirdingASIA* 16: 89-93) and a newly published study on the migration behaviour of far eastern curlews (Morrick et al. 2022 *Conservation Science and Practise* 4: e594), which, to our knowledge are the only studies to date that have addressed this issue. Both studies show that individuals from temperate, southern Australia breed on further **south** (i.e. closer to the equator) than conspecifics from tropical northern Australia. In the case of far eastern curlews, populations from northern and southern Australia therefore breed in relatively close proximity, and latitudinal patterns in the morphology of far eastern curlews at the northern hemisphere breeding grounds likely go against Allen and Bergmann's rule (since far eastern curlews from southern Australia have longer bills and smaller bodies than northern Australian conspecifics). Nevertheless, this is just a single study on a single species and caution is warranted. We have therefore updated our wording in lines 444-456 of the discussion to better acknowledge possible effects of spatial segregation at the breeding grounds.*

2) I'm not sure that I completely understand the rationale for assuming wing length and body mass will change in lock step. While wing length may often be used as an index for body size, and I agree that there are limitations associated with mass, there is also recent work in North America demonstrating that temperature drives reductions in body size, but not wing length. Thus, it seems at least possible – if not likely – that wing length and body size might have contrasting relationships with temperature. At the very least some consideration of this seems necessary.

*As pointed out by the Reviewer, wing length is often used as an index of body size due to limitations associated with body mass (for example body mass is more changeable according to the seasons, among other things). Across our dataset, wing length and body mass are highly correlated ($r = 0.82$) and past research shows wing length and body size scale linearly (Sullivan et al. 2019 *Science Advances* 5: eaat4269). We now include reference to this past research (line 182).*

*However, we agree with the Reviewer that other selective forces – which vary latitudinally – might impact wing length independent of body size (e.g. migration distance as shown in Weeks et al. 2020 *Ecology Letters*). We have updated the wording in lines 185-188 to clarify that wing length may vary latitudinally because of latitudinal differences in migration distance and predation risk, as well as latitudinal effects on body size.*

3) I'm not entirely sure I understand how some of the categories in Fig 1 relate to each other. For example, I assume that relative bill length is bill length relative to mass? If so, couldn't reduced body size drive increases in relative bill length? So for example, the predation risk hypothesis would drive reductions in mass and as a result if bill is constrained by foraging behavior, relative bill length would be expected to increase across all of the categories of migration? This seems similar for the starvation risk hypothesis; nomadic species are not predicted to change in size under this hypothesis, but it is not clear to me why that is. In other words, it seems that at least 3 or 4 of the hypotheses may actually generate the same set of predictions?

We agree with the Reviewer that latitudinal differences in body size might impact relative bill length. To address this, we had assessed latitudinal differences in bill length relative to body size (controlling for wing length) as well as absolute differences in bill length (without controlling for body size effects). However, we had not made this clear in our original manuscript and have therefore updated the opening paragraph of the Methods section to address this, more clearly reflecting the issue raised by the reviewer here (see lines 188-195). The overall patterns in bill length (larger in northern populations) still hold when we consider absolute bill length (see supplementary material Table S3).

We felt it was important to show latitudinal differences in bill size while controlling for individual variation in body size to test whether birds have proportionally longer bills in northern Australia (e.g. leading to a larger unfeathered surface area for heat exchange relative to the bird's body independent of body size variation, for example due to age or sex). We agree that it can be difficult to disentangle the extent to which selection for better thermoregulation might driving smaller body size rather than larger bill size, but we think that by also demonstrating the same patterns in absolute bill length we provide compelling evidence to show changes are not simply due to changes in body size combined with constraints on bill size.

We have changed the heading of Figure 1 from 'Relative bill length' to 'Bill length' to avoid confusion, since the predictions may apply regardless of whether we control for individual variation in body size. We have also included an explanation of why there are no predicted effects of latitudinal trends in starvation risk on nomadic species (nomadic species often move to new locations in search of food; lines 146-147).

4) I'm also not sure that body mass is necessarily a good estimate of the surface area to mass ratio that matters for birds; given that essentially all heat exchange occurs from the appendages, isn't the ratio of wing/bill length to mass a more meaningful surface area:mass ratio?

We think the Reviewer is referring to line 50 where we state "smaller bodies and longer appendages are advantageous in warm climates due to the increased relative surface area available for heat loss". We would have to respectfully disagree with the Reviewer's statement above. Actual heat loss estimates across the body surfaces have been calculated from physical modelling of surface temperature data in various bird species. These show that the body and head actually contribute to the majority of total heat loss. The appendages and bill are important (and relative to their surface area disproportionately so), but certainly not responsible for all, or even the majority, of the body heat exchange with the environment (Tattersall et al. 2018 Functional Ecology 32: 358-368). We have updated the opening section of the Methods to clarify this (line 185).

Again, I greatly enjoyed reading this manuscript, and I think this is likely to be a broadly important study that is building on a long history of amazing work to come out of this group.

Thank you for your helpful criticism and positive feedback!

Reviewer #2 (Remarks to the Author):

This is the first time in my career that I have ever seen a presumed first-submission manuscript and thought it ready for acceptance. Congratulations to the authors – this is a lovely piece of science.

This manuscript considers two ecogeographic rules (Bergmann's and Allen's rules, that species at higher latitudes will be, respectively, larger-bodied and smaller-limbed) in 30 species of shorebirds using a stagger 20,000 datapoints across 46 years. The authors consider six possible evolutionary drivers of these patterns, and determine of those that selection for thermoregulation best explains the morphological differences observed in this dataset. These six hypotheses are very clearly explained and well-grounded in the literature; the models used to demonstrate the statistical relationships are appropriate and well-documented. Ecogeographical rules seem to be making something of a comeback recently, and this is a very elegant investigation of a well-selected system. I could see this study featuring in a lot of textbooks.

(Also, as a small side-note, I appreciate the use of the phrase “community scientists”.)

Thank you for your positive feedback!

My one and only quibble is the apparent assumption that plasticity is not adaptive. This position obviously exists in the field evolutionary biology, but it is controversial, and I would worry that it would detract from the main study used here. The two places I noticed it most were L 28 and L 374-376; these could be easily rephrased if the authors wish to not wade into this particular fight. (For example, in the former, “...improved thermoregulation including plastic responses to...”; in the latter, “driven by population-level genetic adaptation” or “driven by adaptation to...In particular, such patterns”?) Alternatively, I would certainly find a clear discussion of whether or not plasticity is adaptive in light of ecogeographical rules to be fascinating, but I would recommend that the authors make it clearer in-text that this is the position they're taking.

We did not intend to imply that plasticity is not adaptive (determining whether or not this is the case is beyond the scope of this study). We have therefore adjusted the wording as suggested by the Reviewer (lines 37 and 430 of the updated manuscript). We have similarly updated the wording in line 71.

Anyway, congratulations to the authors – I really enjoyed reading this manuscript.

Thanks!

Reviewer #3 (Remarks to the Author):

The present study investigates two historical issues in biogeography, Allen's and Bergmann's rules, using many species of Australian shorebirds as study system and tested patterns of variation in bill and body size at both within-species and cross-species levels. The authors did an unprecedented effort to collect a huge number of individuals of 30 species (the total sample size is >200'000 individuals measured) and compared bill length, body size and body mass between populations located in hot, tropical northern Australia vs. those located in temperate, southern Australia. Most of the analysed species showed pattern of variation compatible with both ecogeographical rules, with populations in the north having longer bills and smaller bodies than conspecifics from the south. The authors also tested for multiple competing hypotheses about the evolution of bill and body size and concluded that, given that similar patterns are observed in species showing different migration and foraging strategies, the most parsimonious mechanism explaining the overall phenotypic variation is thermoregulation (i.e. smaller bills and larger bodies increase insulation in colder climates, and vice versa). The study is interesting, the manuscript is well written, the predictions are clearly stated, and the results are clear. Also considering the large taxonomic and geographic coverage of the species included in the analyses, I think that the study has merits that makes it publishable. However, there are some issues that I would like to raise below in order to improve the quality and the readability of the paper.

We thank the Reviewer for their helpful feedback and address their concerns below. Please note that line numbers refer to the version of the manuscript showing track changes.

1) My main comment concerns the interpretation of the results. Indeed, the study unequivocally shows that Allen's and Bergmann's rules apply to this various group of birds, irrespective of the ecological features of each species. The fact that species showing different migration and feeding strategies show coherent pattern of geographic variation in phenotypic traits is very important because it shows the generality of these patterns. However, I would be more careful in the interpretation of the mechanism(s) behind the observed variation. Indeed, all the competing hypotheses proposed to explain the spatial variation in bill and body size have predictions in the same direction (i.e. increase in body size and decrease in bill length in colder climates) and most of them also predict the same (or similar) strength of the effect for the same category of birds (e.g. resident species). Given the nature of the data used, which are necessarily correlative, the study cannot test which of the competing hypotheses is best supported. On the whole, I could agree with the authors that thermoregulation seems to be the most parsimonious mechanism explaining the overall variation. However, with the present data it is not possible to exclude that different mechanisms can explain variation in bill and/or body size in some categories of birds (e.g. it is possible that phenotypic plasticity during growth explains variation in bill length and body size in resident species more than thermoregulation). Or even that multiple mechanisms act together to produce the observed patterns (e.g. phenotypic plasticity, thermoregulation and starvation risk may explain variation in body size in resident species). The authors used a quite cautious wording in the title and in the very last paragraph of the Discussion, and I agree with such an approach. However, I think that the entire Discussion should be more clear about such a possible multiple concomitant effects, and I am not sure that the use of subheading helps the understanding of the possible complex patterns of variation, with multiple selective forces potentially acting together.

We have updated the wording of our discussion and in Figure 1 to better acknowledge that there may be multiple explanations that, when combined, explain latitudinal patterns in bill and body size

across the different shorebird species (please see updated wording in the opening paragraph of the discussion, lines 394-395; main body of the discussion, lines 444-456, 504-506, conclusion lines 539-540 and Figure 1 caption, lines 139-140). We agree that there are likely multiple drivers of latitudinal patterns in body size and bill length and feel the updated manuscript, combined with the parts where we had already been cautious and qualifying in our explanation (as the reviewer indicates) better acknowledges this. We prefer to keep the subheadings to help the reader relate our discussion back to the results and predictions outlined in Figure 1.

2) A second main point concerns how the authors treated their original data. I think the authors should make more effort to describe their original dataset and provide some additional information that are very relevant for the reader to understand if data are reliable and eventually if the procedures used for data collection may have somehow affected their conclusions. I think that the authors should easily be able to provide all the relevant information that I briefly summarize below.

Thank you for this comment; please see our responses to the detailed feedback below.

2a. As clearly stated by the authors, their database may suffer from the fact that measures were taken by multiple measurers, potentially posing problems in the reliability of the data used. Because a small fraction of the individuals have been captured multiple times, I would suggest to provide some analyses of repeatability of the measurements taken by different measurers. Even if recaptures represent a very small fraction of the individuals, considering the very large capture effort, the authors could rely on an adequate sample of individuals to perform these analyses and therefore to show that their data are solid and results are robust. In addition, I would also add that an understanding of measurement error would help to better interpret the biological relevancy of the results obtained, especially because the relative difference in the analysed traits between southern and northern populations is far from being large (< 2% for bill and wing length). It would therefore be important to know whether measurement error is within the range of the reported spatial variation, but even if it is not (hopefully), it would be important for the readers to know this.

We thank the reviewer for this suggestion. We have now calculated repeatability of measurements within the same individuals following Nakagawa and Schielzeth (2010) by dividing variance explained by individuals (band number) by the residual variance and individual variance combined. Variances were estimated for recaptured individuals (repeated band numbers) using a linear mixed model with bill length, wing length or mass as the response variable, sample year, age (adult/juvenile) and season (summer-wet vs winter-dry) as fixed terms, and species and band ID as random effects.

Repeatability for bill length was high = 0.92. It was lower for wing length = 0.72. In both cases N = 3317 samples of 1583 individual shorebirds (band numbers) captured in the same year. Repeatability for mass was also lower = 0.78, for N = 3168 samples of 1567 individuals captured in the same month and year.

We now include this information in the supplementary material, where we also provide further comprehensive details of the data cleaning methods we employed to manage errors in the dataset.

We note that lower repeatability of wing length and mass are to be expected, as measurements are likely influenced by factors such as recent foraging success prior to capture and feather wear and tear, whereas bill lengths, though impacted by growth and wear, are expected to remain relatively stable. Indeed the high repeatability for bill length for the same specimens, suggests that the lower repeatability for mass and wing length are not predominantly due to observer error.

It should be noted that the shorebirds have been regularly sampled at both northern and southern sites by the **same** small group of community scientists, using consistent methodology and under similar conditions (e.g. caught regularly at low tide). So there is minimal chance of the patterns we identify be the result of consistent differences in measurement protocol between northern and southern sites.

While it is possible that repeatability scores may underestimate true measurement accuracy due to errors in reading band numbers, measurement error will most likely be adding noise to the dataset rather than systematic bias; To this end, the huge number of data points collected over 46 years should mean that our results mitigate the effects of measurement error.

2b. I think that some information is needed also about the consistency of wintering grounds along the life of single individuals of migratory species. In practice, do the same individuals always winter in the same area? Or is there any evidence that some individuals (at least in some species) can overwinter on the northern coast in some years and in the southern one in the other years? This is a crucial point to understand if the premises of the study are met for (all or most of) the migratory species. I think that this information can be extracted again from multiple captures of the same individuals in different years.

This is correct; the same individual shorebirds typically return to the same non-breeding grounds in either northern or southern Australia. Based on recapture data, only 239 of 60,007 (0.4%) individuals have been caught in both northern and southern Australia (now included in lines 217-221).

2c. Another quite tricky point concern the year of capture of different individuals. Although the authors correctly included year of capture as a covariate in the models, I was wondering if the distribution of captures in the two locations is constant over time. Indeed, if thermoregulation is the main cause of geographic variation in these traits, we should expect a variation in bill size (increase), body size (decrease) and body mass (decrease) over time due to the increase in temperatures as a consequence of global warming. I note that trends compatible with this idea, although statistically non-significant (except for body mass), are presented in the results (Tables S2-5). An uneven temporal distribution of the captures may have therefore affected the results (e.g. the observed patterns should have been obtained if a larger capture effort was done in recent years in the north). Thus, some information about this should be included.

Capture effort in northern and southern Australia has been comparable over time. To address this concern, we now show a histogram of sample collection over the years in the supplementary material (see Figure S1).

3) A final consideration concerns statistical analyses. Let me start by saying that statistics are state of the art, and therefore I do not find any major flaw about how the models have been performed and about the packages used. However, it is not clear to me why the analyses of bill variation including interaction terms (i.e. location*migration and location*foraging) were treated in separate models. In my opinion, they should be included in a single model containing multiple interaction terms. I would encourage the authors to provide a single full model or, at least, to justify why it is better to split the analyses.

We analysed the two interactions in separate models because the more complex model with both interactions did not converge. We now include this justification in lines 345-346.

Minor comments:

LL 73-76. It might be worthy to mention that a positive covariation between the size of the predators and that of their prey should also be expected. There is a large body of evidence about such an expected trend, including a few empirical studies showing within-species variation in body size according to spatial variation in the size of the prey. If the authors do not intend to add this one among the competing hypothesis on variation in body size, I think that they should at least acknowledge the reader about it.

We have updated the introduction to acknowledge possible effects of co-variation in body size between predators and prey (lines 86-87), and have inserted some relevant references for this point.

Fig. 3. I think that the figure would benefit from the inclusion of average effect sizes resulting from the across-species analyses per migration category and one indicating the overall effect on all the species used. This would help researchers to compare biological effects across different studies.

We now include average standardised effects for migratory, nomadic and partly migratory, and resident species in the caption of Figure 3. We include the overall effects across all species from phylogenetically controlled analyses as a percentage in the main text (lines 348-361).

LL 378-381. This is not necessary true, at least for some of species. The assumption that all the populations of all the species breed under similar climatic conditions can be questioned, as many species typically show a large breeding area covering a large latitudinal range. This could be a potential problem for the species showing a large migratory connectivity (the spatial and temporal linkages of individuals and populations between seasons that result from migratory movements). Do the authors have some evidence that (some of) the species included in the analyses display a large migratory connectivity with populations breeding in warmer climates also wintering in warmer areas? However, some discussion about this point is recommended.

We have updated our wording in lines 444-456 of the discussion to better acknowledge possible effects of migratory connectivity as suggested by the Reviewer, including reference to a recently published study on the migratory behaviour of far eastern curlews (Morrick et al. 2022 Conservation Science and Practise 4: e564) and a slightly older study of red knots (Hassell et al. 2011 BirdingASIA 16, 89-93). To our knowledge, these are the only studies to date that show any migratory connectivity for a shorebird species included in our study. However, the patterns found at the breeding grounds likely go against Allen and Bergmann's Rule; the Morrnick et al. study shows that far eastern curlews from temperate, southern Australia breed on average 4° further south than conspecifics from tropical northern Australia (far eastern curlews from southern Australia have longer bills and smaller bodies than northern Australian conspecifics). The Hassell study shows a similar pattern for red knots (although red knots do not exhibit any latitudinal differences in bill or wing length. However, these are only two studies, and little is known about the migratory movements of shorebirds that spend their non-breeding seasons in northern and southern Australia; we agree further study of migratory connectivity of populations from the two regions would be insightful.

LL 428-430. I do not fully understand this sentence. Indeed, according to Fig. 1, both the starvation and predation risk hypotheses do not have any prediction about variation in bill length. We cannot exclude that body size would vary according to either or both of these hypotheses and bill length varies according to thermoregulation. This comment is fully related to my first main comment.

Apologies for the confusion as our original wording was not clear. The Reviewer is correct – the starvation and predation risk hypotheses do not make any prediction for bill size – what we meant was that an additional explanation is needed for the observed latitudinal changes in bill length. This does not mean that the starvation and predation risk hypotheses can be ruled out, but makes them a less parsimonious explanation for our results than the thermoregulatory hypothesis. We have re-worded this section and the caption of figure 1 to make this point clear (see updated lines 143-144; 504-506).

Effect of year on body mass. I am surprised that this significant effect, as shown in Table S5, is not discussed at all. I think it is very important to include some interpretations of such an effect in the Discussion, also because it seems to be in line with the interpretation that thermoregulation is the main driver of this trait.

It is indeed interesting that sampling year has a negative effect on shorebird body mass. As the Reviewer points out, this could be driven by increasing temperatures over time and may support our interpretation that selection for thermoregulation is an important driver of latitudinal differences in morphology. However, we do not include this interpretation in our discussion, as testing a year effect is beyond the original scope of our study and we do not also show that temperatures have increased at the study sites over the time period the data was collected, which to our mind makes interpretation of a year effect overly speculative. A thorough investigation of whether shorebirds are becoming smaller in response to warming temperatures at the non-breeding grounds is needed and we are currently working on this as the subject of a separate study.

Reviewers' Comments:

Reviewer #1:

Remarks to the Author:

I think the authors have done a wonderful and thorough job responding to my initial concerns. In all instances in which the authors took issue with my comments, I thought the explanations were thorough and convincing. I see no reason that this manuscript should not be published in its current form, and congratulate the authors on a wonderful piece of work - I think it is a wonderful contribution to the field and is sure to be of broad interest.

Reviewer #3:

Remarks to the Author:

Thank you for the revision and for your extended response to my previous comments. I have a few comments on your replies.

- In the reply to my point 2b, the authors state: This is correct; the same individual shorebirds typically return to the same non-breeding grounds in either northern or southern Australia. Based on recapture data, only 239 of 60,007 (0.4%) individuals have been caught in both northern and southern Australia (now included in lines 217-221).

Thanks for the inclusion of this information. However, it is not clear (at least to me) whether this very low percentage concerns RECAPTURES OF THE SAME INDIVIDUALS IN DIFFERENT YEARS. If yes, please explicit it. If not, please update the analyses on this subsample of individuals. In addition, I am also whether there is a difference among species (i.e. some species very consistent in wintering grounds and other ones quite variable).

- In the reply to my first minor comment, the authors state We have updated the introduction to acknowledge possible effects of co-variation in body size between predators and prey (lines 86-87), and have inserted some relevant references for this point.

You added these references concerning the 'starvation risk hypothesis". However, the trend in body size predicted by the starvation risk hypothesis is the same as predicted by the Bergmann's rule. There are cases of within-species geographic variation in body size of predators according to spatial variation in their prey size. This can be the case even though spatial variation in predator body size does not follow the Bergmann's rule (e.g. populations living in the Tropics can be larger than those living at higher latitudes). I thus suggest adding some considerations (and proper references) about this.

Response to Reviewers:

We thank Reviewer 3 for their final comments and requests for clarification. Our response is as follows:

R3: - In the reply to my point 2b, the authors state: This is correct; the same individual shorebirds typically return to the same non-breeding grounds in either northern or southern Australia. Based on recapture data, only 239 of 60,007 (0.4%) individuals have been caught in both northern and southern Australia (now included in lines 217-221).

Thanks for the inclusion of this information. However, it is not clear (at least to me) whether this very low percentage concerns RECAPTURES OF THE SAME INDIVIDUALS IN DIFFERENT YEARS. If yes, please explicit it. If not, please update the analyses on this subsample of individuals. In addition, I am also whether there is a difference among species (i.e. some species very consistent in wintering grounds and other ones quite variable).

We have amended the text further to clarify this point, and have updated the numbers to more specifically reflect data for migratory species (lines 291-297):

“according to the shorebird dataset used here, 139 of 43,560 (0.3%) migratory individuals recaptured across different years were caught in both northern and southern Australia, possibly due to a change in non-breeding grounds across different years, mistakes when recording band numbers, or because of a stopover in northern Australia on migration from or to southern Australia. Individuals recaptured in both northern and southern Australia across different years include bar-tailed godwits, curlew sandpipers, great knots, red-necked stints, red knots, ruddy turnstones and sanderlings.”

- In the reply to my first minor comment, the authors state We have updated the introduction to acknowledge possible effects of co-variation in body size between predators and prey (lines 86-87), and have inserted some relevant references for this point.

You added these references concerning the ‘starvation risk hypothesis’. However, the trend in body size predicted by the starvation risk hypothesis is the same as predicted by the Bergmann’s rule. There are cases of within-species geographic variation in body size of predators according to spatial variation in their prey size. This can be the case even though spatial variation in predator body size does not follow the Bergmann’s rule (e.g. populations living in the

Tropics can be larger than those living at higher latitudes). I thus suggest adding some considerations (and proper references) about this.

We appreciate the point being made here: there can be patterns in predator body size that run counter to Bergmann's rule, if prey size follows counter trends. However, in the context of the introduction, our focus is specifically on alternative explanations for Bergmann's and Allen's rule patterns. We therefore think further discussion of prey/predator size relationships is too tangential to the paper. The references we cite include general large-scale studies of predator-prey size relationship so seem to us sufficient. However, we have slightly modified the text at lines 69-70, to reflect the more general point that the reviewer is making here:

*"larger animals may also be favoured at high latitudes **if they** consume larger prey or a greater variety of prey sizes ('starvation risk hypothesis'³⁹⁻⁴³)"*